# Touching a Mechanical Body: The Role of Anthropomorphic Framing in Physiological Arousal When Touching a Robot

**DOI:** 10.3390/s23135954

**Published:** 2023-06-27

**Authors:** Konrad Maj, Paulina Grzybowicz, Wiktoria Laura Drela, Michał Olszanowski

**Affiliations:** Faculty of Psychology, SWPS University, 03-815 Warsaw, Poland; pgrzybowicz@swps.edu.pl (P.G.); drelawiktorialaura@gmail.com (W.L.D.); molszanowski@swps.edu.pl (M.O.)

**Keywords:** human–robot interaction, touch, physiological arousal, anthropomorphic framing

## Abstract

The growing prevalence of social robots in various fields necessitates a deeper understanding of touch in Human–Robot Interaction (HRI). This study investigates how human-initiated touch influences physiological responses during interactions with robots, considering factors such as anthropomorphic framing of robot body parts and attributed gender. Two types of anthropomorphic framings are applied: the use of anatomical body part names and assignment of male or female gender to the robot. Higher physiological arousal was observed when touching less accessible body parts than when touching more accessible body parts in both conditions. Results also indicate that using anatomical names intensifies arousal compared to the control condition. Additionally, touching the male robot resulted in higher arousal in all participants, especially when anatomical body part names were used. This study contributes to the understanding of how anthropomorphic framing and gender impact physiological arousal in touch interactions with social robots, offering valuable insights for social robotics development.

## 1. Introduction

Understanding human–robot interaction (HRI) has become increasingly important with the rise in development and use of social robots in fields such as education, elderly care, and therapy [1,2,3,4]. There is a large body of existing studies looking into how humans interact with and perceive social and humanoid robots, and specifically, to what extent such interactions are comparable to human–human interactions (HHI). Touch, a common form of interpersonal communication between humans, is understudied within HRI. However, studies show people regularly seek to engage with robots through touch [5,6,7] and that robot touch may have similar influences to touch between humans [8,9,10]. Understanding the role of touch in HRI, including its similarities to HHI and its social and physiological impact on humans, is essential to the future development and implementation of social robotics, ultimately improving the overall quality and effectiveness of human–robot interactions.

We aim to contribute to this process by exploring the physiological effect of human-initiated robot touch, and further, the influence of anthropomorphic framing on this interaction.

Touch is defined as physical contact between two or more individuals [11], and it has historically served both communicative and relational functions for humanity. Touch allows for the nonverbal communication of messages and emotions, as well as the creation of intimacy and trust between individuals [11]. It has been shown to influence the trust and liking of others [12] and encourage or discourage different prosocial behaviors and performances [13]. Touch between humans also has significant physiological effects: in his arousal model of interpersonal intimacy, Patterson explains how touch evokes measurable changes in physiological arousal [14]. Touch in social communication, for example, in the form of handshaking and hugging, has also proven to reduce stress and anxiety [15,16,17,18] and produce a positive care effect [19].

The physiological responses experienced by humans can be referred to as physiological arousal. This kind of arousal can be triggered by various positive and negative experiences including stress, fear, discomfort, excitement, and even exercise. [20,21] Responses can involve heightened blood pressure, increased heart rate, elevated rates of respiration, and increased perspiration [20,22]

These responses to touch are highly dependent on the social context of the interaction, including which body part is touched. The concept of body accessibility addresses people’s willingness to let others touch various regions of their body [20]. The term was coined by S.M Jourard, who assessed body accessibility according to how frequently people touched and were touched in 24 different regions. While touching hands and arms during handshakes or hugs is generally more acceptable, touching more vulnerable areas, such as the head, neck, torso, lower back, buttocks, and genitalia can be seen as less positive, and even an invasion of privacy, depending on the relationship between individuals [23,24].

Gender dynamics have also been proven to influence touch behavior, which is unsurprising considering the prevalent role of gender norms in society. In their study, Richard Hesin et al. found that women often derive meaning from touch based on their relationship with the other individual, while men tend to be more impacted by the other person’s gender [25]. For example, women in this study found touch from an opposite sex stranger unpleasant and touch from an opposite sex close friend pleasant. On the other hand, male participants were just as comfortable with touch from a woman who is a stranger and a woman who is a close friend. A study conducted by Hubbard et al. found that cross-gender touch results in more favorable perceptions and reactions while waitressing, or counseling [26]. At the same time, it has been shown that men tend to perceive physical contact more positively than women [27].

How much of what is understood about touch transfers to robots? Social robots have already been implemented in various fields, and thus we know that touching social, pet-like robots such as PARO reduces pain and stress [9,10], and robotic arms performing touches can enhance positive emotional responses from human participants [19]. Humans touch robots in similar ways they would touch humans, as shown in a study conducted by Andreasson et al. on affective touch in HRI [28], in which tactile conveyance of emotions on humanoid robot NAO was observed, as well as in a study conducted by Yohanan et al. on how humans touched a haptic creature robot, in which humans were found to convey nine different emotions through touch [8]. In their study, Zhou et al. found that bidirectional social touch (both touching and being touched by a humanoid robot) affected human physiology and led to similar physiological activation patterns [29]. Such studies suggest a similarity in the use and impact of human–human and human–robot touch interactions, but more studies are needed to understand the true extent of this claim, including an exploration of which variables impact the effects of touch within HRI.

One of the more significant studies in this area was conducted by Jamy Jue Li et al. [30]. It explores the physiological impact of touching robots on humans by examining body accessibility. In their study, they instructed participants to touch or point to various body parts of a humanoid robot, varying in levels of accessibility. Their skin conductance response was recorded in order to measure physiological arousal. They found that touching less accessible regions (e.g., genitals, thigh, buttocks) resulted in a higher physiological arousal of human participants compared to more accessible regions (e.g., shoulder, arm, hand). Pointing to these same regions, however, did not result in differences in arousal. The study demonstrates the physiological impact of HRI touch and showcases the transfer of accessibility zones to robots.

The results of this study raise a number of questions about the physiological arousal effect found from touching robots. It is not clear how much of the physiological arousal observed in the study is due to perceiving the robot’s body as human-like and how much of it is due to the semantics used to describe the robot’s body, in this case, human anatomical body part names. Does the observed process of anthropomorphization take place through what we see? Or rather, through the words we hear?

Therefore, the study presented in this paper is a replica of the robot touch study conducted by Li et al., with the addition of several changes and conditions. We wish to study the effect of anthropomorphization on physiological arousal, through the anthropomorphic framing of a robot’s body parts (anatomical names vs. numbered body parts) and gender (male-robot vs. female-robot). Additionally, we conducted the experiment on the humanoid robot Pepper instead of Nao.

Anthropomorphization is defined as attributing human qualities, characteristics, and behaviors to nonhuman entities. Several variables have been proven to influence human anthropomorphization of robots, namely, robot characteristics including physical embodiment [31], movements, gestures [32], and language [33]. Anthropomorphization can also be impacted by humans’ mental models of robots, an internal representation that dictates how we perceive and understand them [34]. This is impacted by our individual experiences and characteristics, such as gender, age [35,36,37,38], and technological experience level [37,38,39], but it can also be altered through the use of language, or linguistic framing.

The way we speak about robots has the power to change our perception and understanding of them. Thus, if we use language to frame robots as if they were human, for example, by introducing one with a human name, backstory, and even gender, we can trigger anthropomorphization. The effects of linguistic framing for the purpose of anthropomorphization, i.e., anthropomorphic framing, have been shown in several studies. Reactions to kicking Spot, a robot dog, were significantly more negative after the dog was given a name and backstory [40]. Similarly, research study participants exhibited more hesitancy when striking the Hexbug Nano, a robotic insect, with a mallet after it was given a name and backstory [41]. In their study, Kopp et al. explored effects of linguistic framing on anthropomorphization and human–robot trust in industrial environments, and found that human-like framing of robots in the workplace increased employee trust when the human–robot relation was perceived as cooperative [42]. Even the pronouns we use when referring to robots have a significant impact: using “he” and “she” instead of “it” can indicate the robot appears to us as a “quasi-other” as opposed to just an object, as Coeckelbergh discusses in his study on the linguistic construction of artificial others [43]. Westlund et al. examined the use of pronouns as well, and found that when experimenters introducing a robot to children spoke to the robot using the personal pronoun “you” instead of the impersonal pronoun “it”, children showed more signs of social interaction with the robot [44].

Attributing gender to the robot through the use of names and pronouns can also itself be a form of anthropomorphic framing, with significant impacts. For example, in their experiment on anthropomorphism in autonomous vehicles, Waytz et al. discovered that providing a name, gender, and voice helped users anthropomorphize the vehicle, and in turn, trust it more [45].

Touch is heavily influenced by context, and thus, it is important to explore the role of framing on touch between humans and robots. To address these limitations of the original study, the robot used in our study were anthropomorphically framed in two ways: the use of anatomical body part names and gendered names and pronouns. During the study, either human body part names or numerical digits were used when instructing participants to touch different regions of a robot. The robot was also attributed either a male or female name (Adam or Ada) and personal pronouns (he, she) during its introduction. The use of this language framed the robot as human-like, impacting the way it is perceived and encouraging anthropomorphization.

We expect to observe the following outcomes:

**H1:** 
*In the condition of anthropomorphically framing the robot body parts through the use of anatomical names, the subjects will feel stronger arousal in comparison to the control condition, in which the parts are referred to using numerical digits.*


The anthropomorphic framing of the robot through use of body part names should increase anthropomorphization, leading to participants utilizing human–human interaction frameworks. Based on what we know about touch and accessibility zones between humans, we hypothesize increased levels of physiological arousal when compared to participants instructed to touch body parts referred to with numbers.

**H2:** 
*Physiological arousal is inversely related to the “availability” of a given part of the robotic body for touch.*


Physiological excitation is defined as the change in electrodermal arousal from the prompt stage to the action stage [46]. Researchers [30] reported differences in skin conductance response when they categorized robot’s bodily parts by their body accessibility rating into high, medium, and low tertiles according to how frequently that region is touched in interpersonal communication, according to Jourard [23]. Thus, with this study, we wish to verify those findings.

Attributing a gender to the robot in our study also allowed us to explore the impact of gender on human–robot touch interactions. In general, there is a well-documented influence of gender in HRI. For example, in their study, Kuchenbrandt et al. showed that the gender typicality of HRI tasks substantially influences human–robot interactions as well as human perception and acceptance of a robot [47]. Further studies have shown that people evaluate a robot of the opposite gender more positively than a same-gender robot [47], and that men tend to trust and engage with female robots more [48].

In regard to anthropomorphization, it seems that men tend to anthropomorphize robots more than women [35] and that they may be more impacted by anthropomorphization. A study conducted by Pelau et al. indicated that men are more sensitive to anthropomorphic characteristics of AI devices [49], and Cheng and Chen’s study indicated that robots with anthropomorphic appearances generate higher pleasure among men in comparison to women [50].

However, there are limited HRI studies on the relationship between gender and touch, and no clear results regarding the role of gender on the physiological arousal of humans when touching robots. When considering the documented gender effect in HRI and the role of gender in touch between humans, there is reason to believe that robot and participant gender will play a significant role in this study and the physiological arousal experienced by participants. It is for this reason that we formulated the following research question:

**RQ1:** 
*How will the physiological arousal experienced from cross-gender touch vary from arousal experienced from same-gender touch?*


Finally, a person’s attitude towards robots has been shown to impact the way they interact with and are impacted by robots. For example, in a study conducted by Cramer et al., it was found that participants’ attitudes towards robots influenced how they perceived human–robot touch interactions: participants with more positive attitudes towards robots found the robots engaging in touch less machine-like [51]. Attitudes were evaluated using the NARS (Negative Attitude towards Robots Scale), developed by Nomura et al. [52]. In their study, Picarra et al. also used this scale to predict future intentions to work with social robots [53]. With this understanding, we wish to further explore how our participant’s attitudes towards robots impacts the physiological arousal they may experience during human–robot touch interaction, and thus we formulated the following research question:

**RQ2:** 
*Will physiological arousal when touching a robot be related to the attitudes and beliefs of the subjects about robots?*


## 2. Materials and Methods

### 2.1. Participants

One hundred and sixty adults were recruited for this study, including eighty females and eighty males. Participants were randomly selected and between 18 and 58 years old. After cleaning the data, 141 participants had valid data: 83 females and 58 males. A majority of participants were university students. All participants consented to participation in the study, and were unaware of the true goal of the study until the experiment was complete and the purpose was clearly explained to them. After the experiment, participants were asked to fill out a questionnaire in which they provided their prior experience with humanoid robots and technology and their general approach and beliefs towards such devices.

### 2.2. Design

A 2 (person-sex: female vs. male) × 2 (robot-sex: robot-female vs. robot-male) × 2 (instruction: body part names vs. digits) between-participants study was conducted in which people were asked to touch a humanoid robot.

### 2.3. Materials

#### 2.3.1. Pepper Humanoid Robot

Unlike the robot touch study conducted by Li et al., in which they used the humanoid robot Nao, the robot used in this study was Pepper from Aldebaran Robotics, owned by the HumanTech Center at SWPS University. This is a humanoid robot standing 1.20 m tall, with an articulated head, eyes, arms, and fingers. The robot does not have a distinct nose, ears, legs, genitals, or buttocks. The robot was programmed by our research team using QiSDK and Android Studio. The application was deployed on the robot and remotely controlled by the experimenter. Pepper also has a 10.1 inch tablet embedded on its chest, which was used to display instructions to participants. Various marked diagrams of the robot were displayed, corresponding with the places the participants should touch the robot (Figure 1).

Though Nao and Pepper are both humanoid robots developed by Aldebaran Robotics, they are physically and functionally distinct. Pepper is a much larger robot compared to Nao: Pepper stands at about 120 cm while Nao is 58 cm tall. Another major difference is related to their body parts. While both robots have clearly articulated heads, eyes, necks, arms and backs, Nao also has distinct legs and feet, while Pepper does not. Instead, Pepper relies on wheels to move around (Figure 2).

#### 2.3.2. Skin Conductance Response Measure and Signal Processing

Electrodermal activity was recorded using a BioPac MP160, digitized with 24-bit resolution, sampled at 1 kHz, and recorded on a PC. All digital transformations and further data extractions were performed with the use of Neurokit2 [56]. The EDA signal was filtered with a 3 Hz cutoff frequency and a 4th order Butterworth filter. Skin Conductance Response was measured as the peak amplitude of the first SCR in each epoch—i.e., the 7 s period when participants were attempting to touch. Since the recorded signal could have contained artifacts due to additional body movements (i.e., loss of balance when touching different parts of the robot, scratching the hands around the electrode placement area, additional movements of the hand with electrodes, body turning, etc.), the recorded videos were analyzed to detect and remove data from such trials.

#### 2.3.3. Final Questionnaire

The final questionnaire used in this study consisted of two parts. The first section collected information about the participant, including questions regarding age, gender, year in school, and direction of studies. Participants were also asked about their dominant hand (left/right) and whether or not they had had any interactions with robots (“Have you ever had personal, direct contact with a humanoid (human-like) robot?”). Next they were asked about their well-being during the experiment in the form of semantic differentials. Participants were prompted to answer “In general, while in contact with the robot I felt …” on a 5-point Likert scale for 5 different pairs of descriptors (bad to good, unnatural to natural, tense to relaxed, threatened to safe, and uncomfortable to comfortable). Cronbach’s alpha coefficient for this scale was named “Feelings during the interaction”: 0.847.

The second section of the questionnaire contained questions from the NARS (Negative Attitudes towards Robots Scale) and BHANU (Belief in Human Nature Uniqueness Scale). Both questionnaires are available in the Polish adaptation [51]. The NARS-PL scale consists of two subscales: the subscale of negative attitudes towards interactions with robots (NATIR) and the subscale of negative attitudes towards robots with human features (NARHT). The questionnaire contains 13 statements such as “I would feel relaxed talking with robots” (NARHT) or “I would feel very nervous just standing in front of a robot” (NATIR). Cronbach’s alpha coefficient for this scale: 0.815.

The BHNU scale consists of 6 questions concerning beliefs about the uniqueness of human nature. Examples of statements include: “A robot will never be considered human” or “A robot will never have morality”. In the case of both tools, respondents responded to the statements on a 5-point Likert scale (1—totally disagree to 5—totally agree). Cronbach’s alpha coefficient for this scale: 0.717.

### 2.4. Procedure

A preliminary test of the experiment was executed before the actual data collection began. The entire procedure was tested three times in a target environment and necessary corrections were added to the software, experimental setup, laboratory setup, and procedure timing.

This experiment took place in a small laboratory room at SWPS University. Prior to beginning, all participants consented to being recorded (without their face being visible). The subject was informed about the course of the experiment. To give credibility and rationality to the experiment, it was presented as “testing the sensitivity of the robot’s sensors to the touch of a human hand”.

The procedure was as follows. A set of Ag/Cl electrodes were placed onto the participant’s fingers, positioned on their non-dominant hand. They were positioned standing in front of the robot, and then instructed to avoid sudden movements, keep around a 30 cm distance from the robot, and not move the hand connected to the measurement device. They were told the robot will display information about where it should be touched on its tablet, which they should follow using the hand not attached to the measurement device. This process took around 5 min. They were then left alone in the room, and the experimentation sequence began.

At the beginning of the experiment, the robot introduced itself to the participant as either a robot-male or a robot-female. This introduction was made up of three components: (1) The experimenter verbally introduced the robot using “he” or “she” pronouns; (2) after the experimenter left the room, the robot presented itself as a woman—“Cześć jestem Ada” (“Hi, I’m Ada”) or man: “Cześć, jestem Adam” (“Hi, I’m Adam”); (3) the voice type used by the robot varied depending on gender, using either a lower-pitched, distinctively male voice or a higher-pitched, distinctively female voice. The gender of the robot (robot-female or robot-male) and the body part labels (anatomical names or digits) were randomly selected for each participant.

The robot then displayed a pre-programmed sequence of body parts on the robot’s embedded tablet. Each participant was asked to touch 11 different places (randomized) three separate times, labeled either with anatomical terminology or numerical digits. Eleven places were used (Figure 3).

The sequence involved a 3 s countdown, followed by a diagram displaying the robot and the body part that should be touched for 7 s, and then a 10 s cooldown at the end. The timing of the sequence was as follows:[[(3 s synchronization + 7 s body part image + 10 s cool down] × 11 body parts] × 3 times (1)

This process took 11 min in total. The body parts shown were randomized for each sub-sequence (see Figure 4).

After the experimental part of the study, participants were invited to a second room where they were asked to complete a final questionnaire on a computer. This process took no longer than 10 min.

Once the questionnaire was complete, participants were informed about the true purpose of the experiment and had the opportunity to ask various questions about the robot and the entire experimental procedure. This debriefing process took approximately 5 min. Furthermore, each participant was rewarded with a book for their participation. In total, the entire procedure took around 30 min.

## 3. Results

In order to conduct analysis on the effect of body availability zones on physiological arousal in participants, all body parts touched were categorized into three accessibility zones: high accessibility, medium accessibility, and low accessibility. The categorizations used in the study were taken from the experiment conducted by Li et al. [30], and are presented in Table 1.

In order to take the sampling hierarchy and handle the missing data, the analysis employed multilevel modeling (MLM) with restricted maximum likelihood performed with the use of Jamovi 2.3 and the gamlj package. The fixed effect’s structure included four a priori selected factors: names of robot parts (body parts vs. numbers), participant gender (female vs. male), robot gender (female vs. male), and the accessibility of robot parts (high vs. medium vs. low—see [30], with high accessibility being the reference level) and their respective interactions, while the random effects structure was selected based on a bottom-up model-building strategy. First, the model was created with a minimal factor structure—i.e., only including random intercepts for participants. Next, random effects of each factor (random slopes) along with their interactions were added to the model. All models that did not fail model convergence were then compared based on Akaike Information Criterion (AIC). The model that fit the data best, except for random intercept, also included random effects of accessibility. The covariance structure was set as correlated (unstructured).

The results show that in general, the group of participants asked to touch robot regions referred to with anatomical body part names were more physiologically aroused than those asked using numerical digits—main effect of names of robot parts B = 0.08 [0.01, 0.15], t(132) = 2.25, *p* = 0.027. Additionally, touching the male robot caused participants to be more aroused than touching the female robot—main effect of robot gender B = 0.08 [0.02, 0.15], t(132) = 2.41, *p* = 0.017. Additionally, we found an interaction between the way robot parts were named and robot gender—B = 0.21 [0.07, 0.64], t(132) = 2.92, *p* = 0.004. A simple effect analysis revealed that, in the case of touching a male robot referred to with anatomical body part names, arousal was significantly increased as compared to naming body parts with numbers (numbers—M = 0.28, SE = 0.04 vs. body parts—M = 0.46, SE = 0.03—B = 0.18 [0.08, 0.28], t(130) = 3.70, *p* < 0.001), but there were no differences in the case of the female robot (numbers—M = 0.30, SE = 0.04 vs. body parts—M = 0.28, SE = 0.03)(see: Figure 5). Finally, accessibility of robot parts also caused differences in participants’ arousal—less accessible parts caused higher arousal (high vs. medium—B = 0.04 [0.01, 0.07], t(692) = 2.19, *p* = 0.029 and high vs. low—B = 0.04 [0.01, 0.07], t(448) = 2.33, *p* = 0.020) (see: Figure 6).

Additional exploratory analyses were conducted to determine whether attitudes toward robots (as measured by NARS and BHNU scales) impacted participants’ reported feelings during the study and their level of physiological arousal. On average, participants’ attitudes toward robots were relatively negative—NARS (M = −0.69, SD = 0.64—the 5-point scale for our calculations was coded from −2 to +2), while their beliefs in human nature uniqueness were slightly positive—M = 0.07, SD = 0.78 (scale between −2 to +2).

Correlation analysis showed a negative relationship between the feelings reported after finishing the experimental procedure and the NARS scale index—Pearsons’ r = −0.30, *p* < 0.001—which suggests that participants with negative attitudes towards robots reported more negative feelings about the interaction they experienced with the robot.

In turn, there was no relationship between the feelings reported by participants after the experiment and the BHNU scale index (Pearsons’ r = 0.03, *p* = 0.699). There were also no significant relationships between the NARS and BHNU scale indices and averaged excitation experienced during experiment (as measured SCR)—respectively Pearsons’ r = −0.07, *p* = 0.407 and Pearsons’ r = −0.06, *p* = 0.467. Importantly, adding NARS and BHNU scale indices as covariates to the tested linear models did not increase model fit.

## 4. Discussion

Our study results contribute to the understanding of touch in human–robot interaction (HRI) by examining the effects of anthropomorphic framing and gender on physiological arousal during touch interactions with robots.

### 4.1. Body Part Availability

In line with our second hypothesis (H2), we found that physiological arousal was inversely related to the “availability” of the robot’s body part, supporting the findings of Li et al. [30]. This indicates that the concept of body accessibility, established in human–human interactions (HHI), can also apply to HRI.

### 4.2. Anthropomorphic Framing of Body Parts

Regarding our first hypothesis (H1), our results demonstrated that anthropomorphic framing of the robot’s body parts through the use of anatomical names leads to stronger physiological arousal compared to the control condition, in which body parts were referred to using numerical digits. The findings suggest that anthropomorphic framing influences touch interactions between humans and robots, supporting previous research on the effects of anthropomorphic framing [31,32,33,40,41,42,43,44]. Anthropomorphic framing may increase the human likeness attributed to robots, leading to participants being more affected by body accessibility zones, which have been shown to increase physiological arousal in human–human interactions. Though our study does not allow us to draw broader conclusions as to the exact reason behind the arousal (discomfort, excitement, novelty), our results do give us reason to believe that anthropomorphic framing of a robot will lead to human participants being more physiologically impacted during the touch interaction between them.

### 4.3. Anthropomorphic Framing of Gender

Our research question (RQ1) explores the impact of gender on physiological arousal during touch interactions with robots. Our results found that both male and female participants were more physiologically aroused when touching the robot-male when compared to touching the robot-female. This could be attributed to the physical embodiment of the robot Pepper, as well as several societal norms regarding cross-gender and same-gender touch present in Western cultures.

Although in our study we attributed gender to Pepper the robot using gendered pronouns and names, it is possible that Pepper’s physical features (shoulder and waist proportions, lack of hair) were perceived as more traditionally male. These physical features may have made Pepper’s male gender attribution more convincing, in turn increasing the physiological impact on participants.

Societal norms surrounding gender also likely play a role in these results. Men engage in less same-sex interpersonal touch than do women [57,58]. Discomfort in same-sex male touch interactions could be due to the fact that men are socialized to restrain emotional expression, especially among other men [57,58,59]. Research shows that men who touch other men are more likely to be perceived as homosexual [60]. Thus, their reluctance to engage in touch could be due to homophobic attitudes and the fear of being perceived as homosexual. This is particularly relevant in our study, as it was conducted in Poland, a country with relatively strong homophobic attitudes [61]. This discomfort men experience during same-sex interactions in HHI could be the reason for male participants’ increased physiological arousal when touching a robot, especially when the robot’s body parts are anthropomorphically framed.

On the other hand, there is reason to believe women may be more physiologically aroused when initiating touch with a male-robot because of the social dynamics of cross-sex touch interactions in HHI. Henley et al. explores the role of power and status in touch and finds that initiators of interpersonal touch are often higher in social status, while recipients of touch tend to be lower in status [62,63]. Henley and Major both found that men initiate touch with women more [59,62]. Female-initiated touch goes against this norm, which could play a role in potential increased discomfort and higher physiological arousal experienced by women in female–male robot touch interactions.

### 4.4. Attitudes towards Robots

The analysis of our final questionnaire showed us that participants’ attitudes towards robots were relatively negative and their belief on human nature uniqueness was slightly positive, but that there was no significant relationship between these NARS and BHNU scale indices and physiological arousal experienced during the experiment. This may be explained by the fact that participant attitude was measured directly after the experimental procedure. Their evaluations were likely strongly influenced by the experiences that took place moments before in the laboratory, and therefore may focus more on their beliefs about Pepper during the experiment instead of their beliefs about robots in general. In the next experiment, it would be worth exploring the importance of attitudes towards robots by having participants complete a questionnaire before they engage in the human–robot touch interaction.

### 4.5. Significance

This exploration of how body accessibility and anthropomorphic framing impact human physiological arousal in robot touch interactions offers valuable insights for social robotic development. Understanding which body parts can make a user uncomfortable or excited is important when designing a humanoid robot’s physical features, and further, when choosing placement of tactile sensors that the user is expected to interact with. In certain settings it may not be necessary to attribute these human-like features to robots, knowing they can generate additional physiological stimulation in people.

Understanding this relationship can also help guide the process of appropriately implementing social robots in different settings, be it education, elderly care, or hospitals. For example, while educational and recreational implementations may benefit from touch that increases physiological arousal, robots in hospital settings will likely want to avoid the arousal created by certain touch interactions. Knowing that gender attribution and anatomical body part names have an effect on the physiological state of participants tells us we must be intentional about how we frame robots in each context they are used, depending on the desired result of the interaction.

### 4.6. Limitations and Future Work

Although the study showcases a strong relationship between anthropomorphic framing of body parts and physiological arousal of participants during human–robot touch interaction, a more thorough exploration of participants’ anthropomorphization levels is necessary in future studies. The effect of anthropomorphic framing on anthropomorphization could be investigated using various methods of measurement such as questionnaires and behavior measures.

Another limitation to our study is the so-called novelty effect, which states that people can respond differently to new technologies than they would from sustained use of said technologies over time [64]. Because of this, in future research it may be worth exploring the physiological impact of robot touch once the novelty effect has worn off. Perhaps then, no physiological stimulation will occur in the subjects, regardless of anthropomorphic framing of body parts and gender.

Our study was conducted in Poland, a country with a moderately touch-friendly culture common in Central and Eastern Europe [65]. Because cultural context is so important for touch, it would be worth exploring the effects of this study in both more (i.e., Latin American countries [66] and Southern European countries [67]) and less (i.e., East Asian countries [68] and Northern European countries [67]) touch-friendly cultures. Because gender dynamics vary in different cultures, the role of touch and gender in HRI could also be further explored by conducting studies in countries with various societal norms surrounding gender (i.e., traditional vs. modern approaches to gender roles).

In general, physiological arousal in HRI should be further studied in future experiments. Possible extensions can include a deeper exploration of the type of physiological arousal experienced, whether it be discomfort, stress, pleasure, or excitement.

Wider inference from the obtained results is also limited due to the specific construction, design of our robot, and the material from which it was made. We purposefully selected Pepper, a larger, less toy-like robot than Nao, to use in this study; however, these types of machines have very different characteristics, and our results may not be replicated when using another robot.

The lack of recent references in this research paper can be attributed to the relatively limited activity within the field of study. These older references remain significant in establishing the basis for understanding the subject matter.

## Figures and Tables

**Figure 1 sensors-23-05954-f001:**
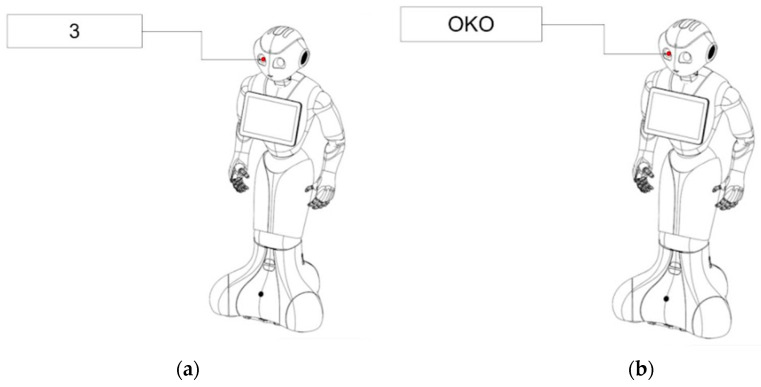
Examples of robot diagrams displayed to participants, labeled with the body part that should be touched, using either a digit (**a**) or the anatomical body name (**b**) (“OKO” means “eye” in polish).

**Figure 2 sensors-23-05954-f002:**
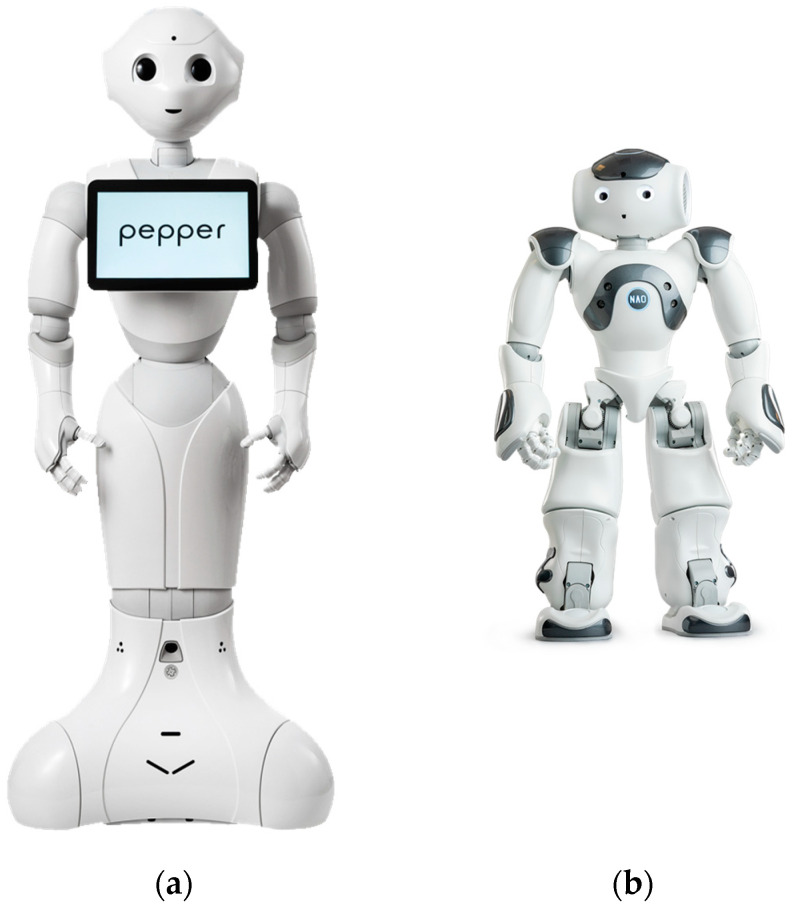
Diagram of humanoid robots (**a**) Pepper [54] and (**b**) Nao [55].

**Figure 3 sensors-23-05954-f003:**
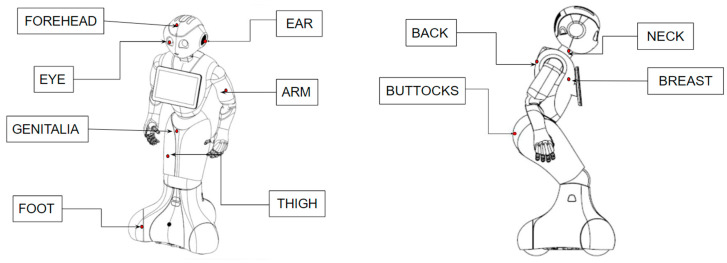
Diagrams displaying the 11 robot body parts that should be touched by participants.

**Figure 4 sensors-23-05954-f004:**
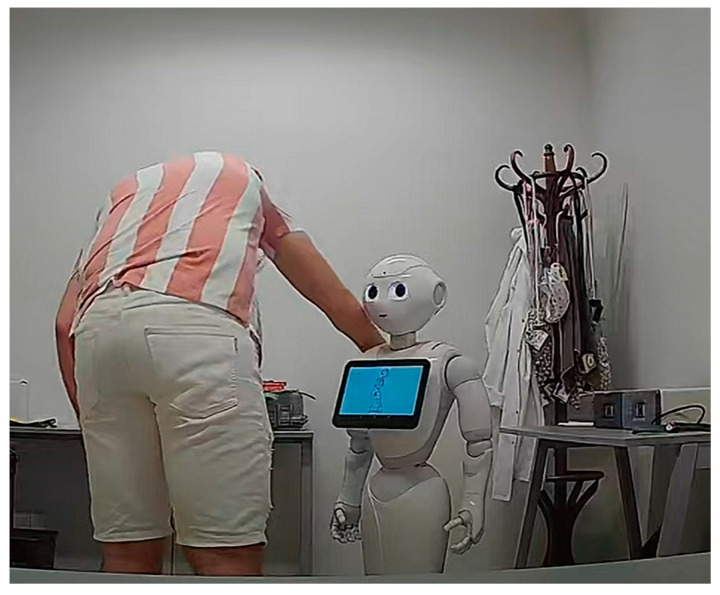
Screenshot from the recording of the procedure. The person is standing next to the robot in a laboratory and touching the robot according to the displayed instructions.

**Figure 5 sensors-23-05954-f005:**
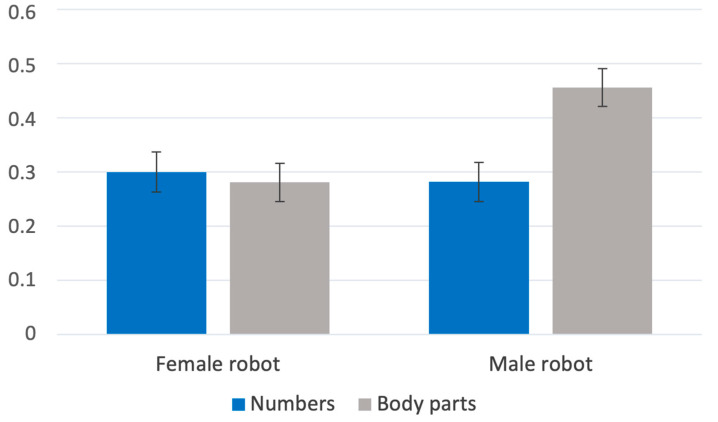
The level of physiological arousal when touching the robot depends on the sex of the subject and the “gender” of the robot.

**Figure 6 sensors-23-05954-f006:**
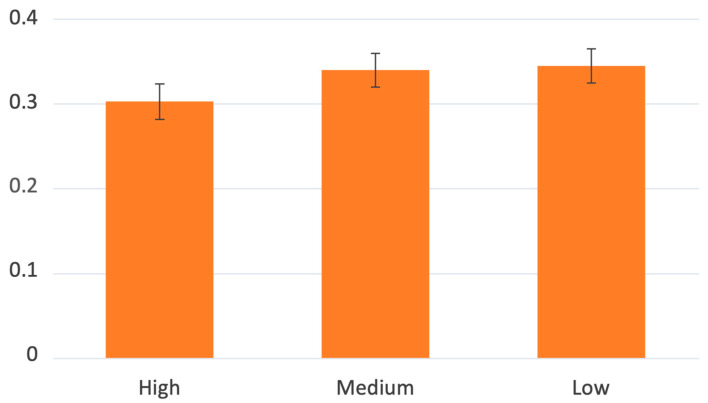
The level of physiological arousal when touching the robot depends on the part’s accessibility.

**Table 1 sensors-23-05954-t001:** Anthropomorphized body parts used in the experiment.

Robot Body Part (Anatomical Term)	Robot Body Part (Numerical Digit)	Accessibility ^1^
Arm	1	High
Arm	7	High
Neck	9	High
Eye	3	Medium
Back	5	Medium
Foot	8	Medium
Ear	10	Medium
Genitalia	2	Low
Breast	4	Low
Buttocks	6	Low
Thigh	11	Low

^1^ Accessibility zone categorization is based on study conducted by Li et al. [30].

## Data Availability

The data presented in this study are openly available in FigShare at https://doi.org/10.6084/m9.figshare.23272214.v1 (accessed on 31 May 2023).

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
