# Peer review of "Touching a Mechanical Body: The Role of Anthropomorphic Framing in Physiological Arousal When Touching a Robot"

_sensors, 2023, doi:10.3390/s23135954_

Round 1

Reviewer 1 Report

The topic addressed by the paper is original and quite interesting. It is well presented, and the hypotheses backed by the results, where the experiments have been conducted in a consistent and sound way. The main concern of this reviewer is about some finer graining on the meaning of arousal, or more precisely, to make the different perceptions of arousal explicit. Arousal is associated to quite different behaviors, such as fight-or-flight response, or sexual activity, and perceived accordingly as uncomfortable or pleasant. This is specially relevant as for RQ1, and commented in part in 4.3, but in my opinion requires some more discussion. The point is that the research seems to be focused on arousal as eliciting uncomfortable feelings, but it could also be quite the contrary, helping to maintain the interest in the particular HRI. Possible future extensions of this research may include experimenting with subjects of more "touch-friendly" cultures, as well as the other way round, humans touched by robots without and with anthropomorphic and gendered traits and/or naming.

Author Response

Dear Reviewer,

Thank you for reviewing our manuscript. Your main point was about the meaning of arousal, or more precisely, to make the different perceptions of arousal explicit. Below are our answers and a description of the changes made:

We have introduced into our paper a more precise meaning of physiological arousal in the introduction section, which reads as follows:

“The physiological responses experienced by humans can be referred to as physiological arousal. This kind of arousal can be triggered by various positive and negative experiences including stress, fear, discomfort, excitement, and even exercise. [20-21] Responses can involve heightened blood pressure, increased heart rate, and elevated rates of respiration, and increased perspiration [20][22]”

We also adjusted some of our wording throughout the article to make clear that physiological arousal can be due to more than just discomfort. Additionally, we added further exploration of physiological arousal in our discussion: 

Anthropomorphic framing may be increasing the human likeness attributed to robots, leading to participants being more affected by body accessibility zones, which have been shown to increase physiological arousal in human-human interactions. Though our study does not allow us to draw broader conclusions as to the exact reason behind the arousal (discomfort, excitement, novelty), our results do give us reason to believe that anthropomorphic framing of a robot will lead to human participants being more physiologically impacted during the touch interaction between them. “

And we discussed possible extensions of the research in our limitations section:

“Our study was conducted in Poland, a country with a moderately touch-friendly culture common in Central and Eastern Europe [65]. Because cultural context is so important in touch, it would be worth it to explore the effects of this study in both more (i.e Latin American countries [66] and Southern European countries [67]) and less (i.e East Asian countries [68] and Northern European countries [67]) touch-friendly cultures. Because gender dynamics vary in different cultures, the role of touch and gender in HRI could also be further explored by conducting studies in countries with various societal norms surrounding gender (i.e traditional vs. modern approaches to gender roles).

In general, physiological arousal in HRI should be further studied in future experiments. Possible extensions can include a deeper exploration of the type of physiological arousal experienced, whether it be discomfort, stress, pleasure, or excitement”

We would like to thank you for your valuable comments and suggestions during the review process of our research. This insight has provided us with new perspectives on our work and allowed us to improve its quality and comprehensibility. We are grateful for your time.

Reviewer 2 Report

This is an interesting study which I believe is not widely explored scientifically. The list below presents my suggestions on how the article can be further improved:

a) Pg 3 P2: From this sentence a reader will appreciate on some discussion or description on anatomical differences between Nao and Pepper. Description in 2.3.1 is inadequate.

Pg4 P10: [48] is an old study. Any newer study with updated questionnaire? Or a justification is required.

Pg5 P3: What is the process of cleaning of data? Do sexual orientation and preferences included in the cleaning up process?

Pg 5 P3: Is the data from this questionnaire relevant as the participants already exposed to the robot whilst doing experiment? 

Pg7 P4: A diagram of the robot body parts to be touched should be included in the paper for ease of discussion.

References: Majority of references are not up to date (more then 5 years). If this is not an active field of research then authors should justify in the background.

Author Response

Dear Reviewer,

Thank you for reviewing our manuscript. Thank you for reviewing our manuscript. Below is a description of the changes made and responses to individual comments.

  • discussion or description on anatomical differences between Nao and Pepper

To address this issue, we have added a detailed description of both Nao and Pepper in the methods and materials section of our article, as well as more information about their differences. We also included a side-by-side comparison using diagrams. 

“Though Nao and Pepper are both humanoid robots developed by Aldebaran Robotics, they are physically and functionally distinct. Pepper is a much larger robot compared to Nao: Pepper stands at about 120 cm while Nao is 58 cm tall. Another major difference is related to their body parts. While both robots have clearly articulated heads, eyes, necks, arms and backs, Nao also has distinct legs and feet, while Pepper does not. Instead, Pepper relies on wheels to move around (Figure 2).”

  • [48] is an old study. Any newer study with an updated questionnaire? Or a justification is required./ References: Majority of references are not up to date (more than 5 years). 

In regards to [48], we chose to use the Negative Attitude Toward Robots Scale partially because the Polish adaptation of this questionnaire was validated in a study by Pochwatko et al. [55]]. Unfortunately, there are not any newer studies with updated questionnaires that we are aware of. 

A majority of our resources are not recent due to a lack of new activity in the field of study, but nonetheless, these HRI studies are significant and important in establishing an understanding of our study. The oldest references are literature on human touch, which remains relevant even now. 

We do mention in our introduction that touch in HRI is understudied. To further address this, we added this justification when discussing our limitations: 

“The lack of recent references in this research paper can be attributed to the relatively limited activity within the field of study. These older references remain significant in establishing the basis for understanding the subject matter.”  

We also added several new resources.

  • What is the process of cleaning of data? Do sexual orientation and preferences included in the cleaning up process? 

We did not take into account sexual orientation and preferences in our study. This is a very intimate question, perhaps some people would not give this information at all, and some would not answer honestly. It is difficult to say what impact the control of these variables would have on the results, and considering that there are only about 5% of non-heteronormative people in Poland, we came to the conclusion that the questionnaire will not contain such questions. Further, we did not anticipate sexual orientation having an effect on our result. In our discussion, we speculate it may explain some of our results. Further exploration of the relationship between sexual orientation and the physiological effects of touching robots can be conducted.

  • Is the data from this questionnaire relevant as the participants already exposed to the robot whilst doing experiment? 

The data collected in the final questionnaire was obviously influenced by the interaction with the robot. However, handing over a questionnaire to complete before interacting with the robot would probably be invasive for robot interaction and arousal measurements. We chose the first solution, although it is worth asking the subjects about their attitudes towards robots before the experiment in future research. And preferably a few days before it.

  • A diagram of the robot body parts to be touched should be included in the paper for ease of discussion.

To address this issue, we added diagrams of Pepper with robot body parts to be touched and labeled (Figure 3). Additionally, we added a table of the body parts (Table 1), as well as a description of the accessibility zones of each body part:

In order to conduct an analysis on the effect of body availability zones on physiological arousal in participants, all body parts touched were categorized into three accessibility zones: high accessibility, medium accessibility, and low accessibility. The categorizations used in the study are taken from the experiment conducted by Li et al. [30], and presented in Table 1.

We would like to thank you for your valuable comments and suggestions during the review process of our research. This insight has provided us with new perspectives on our work and allowed us to improve its quality and comprehensibility. We are grateful for your time.